# 2,5-Diketo-D-Gluconate Hyperproducing *Gluconobacter sphaericus* SJF2-1 with Reporting Multiple Genes Encoding the Membrane-Associated Flavoprotein-Cytochrome c Complexed Dehydrogenases

**DOI:** 10.3390/microorganisms10112130

**Published:** 2022-10-27

**Authors:** Haelim Son, Sang-Uk Han, Kyoung Lee

**Affiliations:** Department of Bio Health Science, Changwon National University, Changwon 51140, Korea

**Keywords:** *Gluconobacter sphaericus*, 2,5-diketo-D-gluconate, membrane-associated flavoprotein-cytochrome c complexed dehydrogenase, L-ascorbic acid, D-glucose biotransformation, D-gluconate

## Abstract

*Gluconobacter sphaericus* has not yet been used in biotransformation studies. In this study, *G. sphaericus* SJF2-1, which produces a diffusible pigment, was isolated from grape. The spent culture medium became dark black when the cells were grown in medium containing glucose and then autoclaved. This bacterium produced 2,5-diketo-D-gluconate (2,5-DKG) from D-glucose and D-gluconate. When 5% D-glucose was used, the conversion efficiency was approximately 52.4% in a flask culture. 2,5-DKG is a precursor of 2-keto-L-gulonic acid, which is a key intermediate in the industrial production of L-ascorbic acid. The complete genome sequence of *G. sphaericus* SJF2-1 was determined for the first time in the *G. sphaericus* species. The total size was 3,198,086 bp, with 2867 protein-coding sequences; one chromosome and six plasmids were identified. From the genome of SJF2-1, multiple genes homologous to those involved in the conversion of D-glucose to 2,5-DKG were identified. In particular, six different genes encoding membrane-associated flavoprotein-cytochrome c complexed dehydrogenase were identified and divided into two different lineages. This study suggests the potential of *G. sphaericus* SJF2-1 to mass-produce 2,5-DKG and other D-glucose oxidation products.

## 1. Introduction

L-ascorbic acid (vitamin C) is an essential nutrient for humans, acts as a coenzyme for collagen synthesis and intracellular energy metabolism, and has an antioxidant effect that removes radicals [1]. The global vitamin C market was valued at 1.55 billion USD in 2021 [2]. 

Currently, L-ascorbic acid is industrially manufactured in six steps, including two fermentation steps, by modifying the previous Reichstein–Grüssner process [3,4]. First, D-glucose is converted to D-sorbitol through a catalytic hydrogenation reaction, and then to L-sorbose using *Gluconobacter oxydans* in the first fermentation step. Next, the culture supernatant is pasteurized and used for the second fermentation process to produce the key intermediate for L-ascorbic acid synthesis, 2-keto-L-gulonic acid (2-KLG), using a mixed culture of *Ketogulonicigenium vulgare* and *Bacillus megaterium* or *Bacillus cereus*. The final chemical conversion stage includes the esterification and lactonization of purified 2-KLG to produce L-ascorbate (Figure 1). The production of 2-KLG has been challenged to improve the current process, including costly hydrogenation, long-term incubation, and additional sterilization steps. One of the challenges is a novel two-step fermentation process that involves the production of 2,5-diketo-D-gluconate (2,5-DKG) from D-glucose by biotransformation and subsequent additional fermentation with a bacterial strain that expresses regiospecific 2,5-DKG reductase to form 2-KLG (Figure 1). One example involves the first fermentation using *Erwinia* sp. for 2,5-DKG production, and the fermentation step with *Corynebacterium* sp. that expresses NADPH-dependent 2,5-DKG reductase. This fermentation showed an 84.6% yield from D-glucose in 10-m^3^ conventional fermenters [5]. A similar example is the production of 2-KLG via a co-immobilized culture of *G. oxydans* ATCC 9937 and *Corynebacterium* sp. using D-gluconate as a starting substrate [6]. The two-stage process was further advanced by designing a single-strain one-step fermentation process using recombinant *Erwinia herbicola*, which expresses the 2,5-DKG reductase gene of *Corynebacterium* [7]. When supplied a total of 40 g/L of D-glucose to the culture medium of the recombinant strain, 49.4% of the D-glucose was converted to 2-KLG during 72-h fermentation [8]. For these one- and two-step processes to be industrially feasible, selecting a high-efficiency bacterial chassis for 2,5-DKG production is a key factor [9].

Some strains of *Erwinia* and *Gluconobacter* have been used in fermentation studies to produce 2,5-DKG from D-glucose [5,7,9,10]. The conversion of D-glucose to 2,5-DKG via 2-KG (2-keto-D-gluconate) by *Gluconobacter* strains is shown in Figure 1. The first three enzymes (E1–E3) are activated in the periplasmic space or on the periplasmic side of the cytoplasmic membrane to promote substrate accessibility and product diffusion from the cell. Through these reactions, the electrons form reduced ubiquinone (coenzyme Q) and are transferred to membrane-bound ubiquinol oxidase, where they reduce O_2_ to form H_2_O and generate energy [11]. In most cases, the activity of these enzymes is not coordinated, and intermediates, such as D-gluconate and 2-KG, often accumulate in the final culture medium [10,12]. This prevents the homogeneous production of 2,5-DKG. When using *G. oxydans* with a high concentration of glucose (10%), the conversion to 2,5-DKG of 66% for 120 h [10] and 21.6% for 60 h [9] have been reported. In the latter case, fed-batch fermentation was performed to increase the yield to 48% at 48 h. *Tatumella citrea* CICC 10802 (formerly *Erwinia citrea*) yielded 23.6 g/L with 10% glucose [9].

*G. sphaericus* NBRC 12467^T^, isolated from grapes, has been reported to produce 2,5-DKG [13,14]. However, the related genes and fermentation properties involved in biotransformation by this strain are not fully known, even though *G. sphaericus* is closely related to *G. oxydans*, which is widely used as a versatile biocatalyst in the manufacturing of L-ascorbic acid, dihydroxyacetone, miglitol, and gluconic acid and its derivatives [15,16]. Here, we present the possibility of using *G. sphaericus* SJF2-1 for the mass production of 2,5-DKG. Moreover, the first complete genome sequence of *G. sphaericus* in this study could further broaden the application range of this species for biotransformation. 

## 2. Materials and Methods

### 2.1. Strain Isolation and Bacteria Culture

The spoiled Campbell’s Early grape collected at Sangju, Korea, was scrubbed on YPD agar (1% yeast extract, 2% peptone, 2% glucose, and 1.5% agar; KisanBio, Seoul, Korea), and a bacterial colony that produced a brown diffusive pigment was selected and designated SJF2-1. This strain has been deposited in the Korean Collection for Type Cultures (KCTC 82305). The strain was routinely cultured in YPS medium (1% yeast extract, 2% peptone, and 2% D-sorbitol), on which the pigment did not produce. API 50 CH (bioMērieux, Craponne, France) was used, according to the manufacturer’s protocol, to test the acid production from 49 carbohydrates. For species identification of the isolate, the 16S rRNA gene was amplified by PCR with primers 27F (5′-AGAGTTTGATCCTGGCTCAG-3′) and 1492R (5′-GGTTACCTTGTTACGACTT-3′), and the nucleotide sequence obtained by Sanger sequencing was used for homology searches in the NCBI database. PCR was performed using a TRIO thermal cycler (Biometra, Goettingen, Germany). The amplification conditions were as follows: (1) at 95 °C for 5 min; (2) 30 cycles of 95 °C for 30 s, 53 °C for 30 s, and 72 °C for 1 min; and (3) 72 °C for 5 min. Sanger sequencing was performed at GenoTech Co. (Daejeon, Korea). As a medium for 2,5-DKG production, D-glucose was separately added to YPP (1% yeast extract, 2% peptone, 0.3% K_2_HPO_4_, 0.25% CaCO_3_, 0.05% MgSO_4_, 0.1% (NH)_2_SO_4_) medium. An amount of 0.1 mL of seed from YPS medium cultured for 36 h was inoculated into a 50 mL medium contained in 250 mL-Erlenmeyer flasks and incubated in a rotary shaking incubator (140 rpm) at 28 °C. The experiment was performed in triplicate. pH was measured using a LAQUAtwin pH meter (Horiba, Kyoto, Japan). Cell growth was monitored by measuring absorbance at 600 nm using a spectrophotometer (nano-MD model, Sinco Co., Seoul, Korea). Strain *Fusarium moniliforme* var. *subglutinans* was obtained from the Korean Collection for Type Cultures (KCTC 6149).

### 2.2. Analyses of Metabolites

Thin layer chromatography (TLC) was performed on plastic-backed silica gel 60 F_254_ (0.1 mm thickness; Sigma-Aldrich) using a mobile phase consisting of ethyl acetate: acetic acid: methanol: deionized water (10:2:2:4, *v*/*v*). Sugar and sugar acids were detected by spraying a color-developing agent (a mixture of 1 g diphenylamine, 1 mL aniline, 50 mL acetone, 7.5 mL phosphoric acid) [17], followed by drying and heating at 120 °C for 5 min. 

For HPLC measurements, a YL9100 HPLC system (Young Lin Instrument Co., Anyang, Korea) with a YL9170 refractive index (RI) detector was used. The ion exclusion column Rezex ROA Organic Acid H+ (8%, 300 × 7.8 mm; Phenomenex, Torrance, CA, USA) was operated with 0.005 N H_2_SO_4_ at 30 °C at a flow rate of 1 mL/min using an isocratic mode for 20 min. However, 2,5-DKG is not commercially available. Because the response of the RI detector is less sensitive to sugar isomers [18], a calibration curve of 2-KG was used to roughly quantify 2,5-DKG. The calibration curve was obtained by plotting the concentrations of 2-KG (0.2%, 0.6%, 0.8%, 1.0%, and 1.2%) versus the area of the respective peaks using the Microsoft Excel program (version 2019). The curve yielded a straight-line relationship, with a coefficient of determination (R^2^) of 0.998. Other products were quantified and compared to those of the standard compounds. Because the D-glucose and D-gluconate peaks overlapped on the ion exclusion column, D-glucose was detected using a ZORBAX carbohydrate column (4.6 mm × 150 mm; Agilent, USA) with an RI detector. The HPLC conditions were set with: mobile phase, 75% ACN in H_2_O; column temperature, 35 °C; injection volume, 10 µL; and flow rate, 1 mL/min for 10 min. 

Mass analysis of the metabolites was carried out using a Q Exactive Plus Quadrupole-Orbitrap Mass Spectrometer (Thermo Scientific, Waltham, MA, USA). The samples were then loaded via direct infusion. The electrospray ionization spray voltage was 2.5 kV in the negative ion mode. The ion transfer capillary temperature was 320 °C, the sheath gas N_2_ pressure was 15 (arbitrary units), the auxiliary gas pressure was 5 (arbitrary units), and the sweep gas pressure was 0. Mass spectra were recorded in the m/z range of 75–1000.

### 2.3. Genome Sequencing and Analyses

The genomic DNA of SJF2-1 was sequenced using the Illumina NovaSeq6000 sequencer (Illumina Technologies, Hayward, CA, USA) and the PacBio RS II platform (PacBio Sciences, Menlo Park, CA, USA) at DNALink Co. (Seoul, Korea). For Illumina sequencing, a TruSeq DNA PCR-free 550-bp library kit (Illumina) was used for library preparation, and the read length was 2 × 151. The quality of the raw sequencing data was verified using the FastQC program (https://www.bioinformatics.babraham.ac.uk/projects/fastqc/) [19] accessed on 20 December 2020. A total of 28,220,814 reads with 8523 Mbp with a mean Phred quality score of 35.22 were generated with a GC content of 57.30%. For PacBio sequencing, a DNA library was prepared using SMRTbell templates with an insert size of 20 kb, as previously described [20]. Raw reads were filtered using a single-molecule real-time (SMRT) portal (version 2.3.0). After subread filtering, 107,194 reads and 8823 Mbp were obtained, with a GC content of 53.95%. The reads were assembled de novo using a hybrid Nanopore-Illumina program in Unicycler version 0.4.9b [21]. 

The quality of the assembled genome sequences was evaluated using CheckM v1.1.3 [22]. Gene predictions and annotations were provided by NCBI using the best-placed reference protein set and GeneMarkS-2+ of the NCBI Prokaryotic Genome Annotation Pipeline 6.1 [23]. The BLAST average nucleotide identity (ANIb) values between the SJF2-1 strain and type species in the database were retrieved from the JSpeciesWS server (https://jspecies.ribohost.com/jspeciesws/) [24] (accessed on 19 August 2022). Digital DNA–DNA hybridization (dDDH) values were calculated by applying the genome-to-genome distance calculator (GGDC 3.0) using formula 2 (https://ggdc.dsmz.de/ggdc.php (accessed on 19 August 2022)) [25].

Genome comparisons based on deduced amino acid sequences were conducted using Rapid Annotations using Subsystems Technology (RAST) [26]. The cumulative GC skew and GC content were depicted using the CGview program with sliding window sizes of 1000-bp and 100-bp steps [27]. The signal peptides and transmembrane domains of proteins were searched for using SignalP-5.0 [28] and TMHMM [29], respectively. The identities between the proteins were calculated by alignment of deduced amino acid sequences using the ClustalW Multiple alignment program (1000 bootstraps of NJ tree) and BLOSUM 62 in BioEdit Alignment Editor software version 7.0.5.3. Phylogenetic trees of proteins were constructed in MEGA_64 using Maximum likelihood analysis with the Jones–Taylor–Thornton model [30]. 

## 3. Results and Discussion

### 3.1. Identification of Strain SJF2-1

Strain SJF2-1 was isolated from spoiled Campbell Early grapes with a very sour taste, as described in Materials and Methods. During incubation on YPD agar for three days, a diffusible light brown pigment was formed by SJF2-1. SJF2-1 was Gram-negative and short-rod (0.8~1.2 µm × ~0.5 µm) under microscopy (Figure 2a,b). 

In the API 50 CH test, SJF2-1 produced acid from L-arabinose, D-xylose, D-galactose, D-glucose, D-mannose, D-melibiose, D-fucose, and potassium 5-keto-D-gluconate (5-KG). The nucleotide sequence of the 16S rRNA gene (1020 bp, NCBI accession number OP077118) showed more than 99.9% similarity to those of *Gluconobacter cerevisiae*, *Gluconobacter albidus*, *Gluconobacter cadivus*, *Gluconobacter sphaericus*, and *Gluconobacter kondonii*. Interestingly, when the 3-d cultured YPD medium was autoclaved, the culture medium changed color to dark black (Figure 2c). The same color was formed even when the pH was adjusted to neutral before autoclaving. This phenomenon was also reported in *Pantoea citrea*, which is known to cause pink disease in pineapples [31]. In this study, the black color was caused by 2,5-DKG produced by *P. citrea*. Furthermore, SJF2-1 inhibited the growth of *Fusarium moniliforme* var. *subglutinans* (Figure 2d). This could be due, in part, to the pH drop to around 2, showing an example of a survival strategy of the 2,5-DKG-producing *G. sphaericus* in competition with the other species in sugar-rich environments. 

### 3.2. Biotransformation of Glucose to 2,5-DKG and Metabolite Analyses

First, the bioconversion pathway and substrate preference involved in the production of 2,5-DKG by SJF2-1 were investigated by TLC and HPLC analyses of culture supernatants after 96 h of incubation. Cells were cultured in the YPP medium with 1% substrates of D-glucose, D-gluconate, 2-KG, and 5-KG.

In TLC analysis, standard chemicals were visualized using a color-developing spray agent with a spot loading of 40 µg, except for a ten times higher amount of D-gluconate for visible color development. In addition, there were differences in the color of each chemical. D-glucose, 2-KG, and 5-KG are blue; 2,5-DKG is deep blue; and D-gluconate is red (Figure 2a). The substrates D-glucose, D-gluconate, and 2-KG produced a single dominant product with the same color. The supernatant of D-glucose was analyzed by LC/MS, and the largest mass peak, with an m/z of 191.02 [M-H^+^], expected for 2,5-DKG, was identified (Figure 2b). Thus, the products produced from D-gluconate and 2-KG were concluded to be 2,5-DKG. When 5-KG was used as the substrate, the color of the product on TLC was different from that of 2,5-DKG. Additionally, we found that the 2,5-DKG spot on TLC disappeared from the black-colored sample produced by autoclaving.

The results of HPLC with the Rezex ROA Organic Acid H+ column in a mix of triplicates are presented in Figure 3c. D-glucose (10 mg/mL) was completely consumed during 96 h of incubation, and one product peak in the HPLC chromatogram appeared at 4.50 min. We considered 2,5-DKG based on TLC and LC/MS results, as described above. The amount of 2,5-DKG was approximately 6.8 mg/mL (68% yield). Under the same incubation conditions, D-gluconate was consumed by 45.2% with an accumulation of 2-KG (1.8 mg/mL) and 2,5-DKG (2.2 mg/mL). 2-KG was less consumed, by 13.8%, yielding 0.07 mg/mL of 2,5-DKG. 5-KG was most consumed, by 89.2%. A product peak was observed at the same retention time as 2,5-DKG, but it was considered to be a different substance from 2,5-DKG by TLC analysis. Accordingly, D-glucose was the most effective substrate for the production of 2,5-DKG by SJF2-1, and D-gluconate was converted to 2,5-DKG through 2-KG, as presented in Figure 1. In contrast, 2-KG and 5-KG are not effective substrates for biotransformation. This result may reflect the differential expression of genes related to 2,5-DKG production by substrates.

The results of biotransformation using 5% D-glucose to investigate the fermentation pattern during the 96-h incubation period are shown in Figure 4. D-glucose, a substrate, was rapidly consumed between 24 and 48 h. The initial high OD_600_ value was attributed to insoluble CaCO_3_, and the OD_600_ values during the first 48-h of incubation were a combination of CaCO_3_ dissolution and increased cell density. D-gluconate, as an intermediate metabolite, accumulated maximally at 48 h, and was relatively slowly converted to 2,5-DKG. 2-KG was not accumulated. Finally, approximately 26.2 mg/mL of 2,5-DKG (52.4% yield) was produced. At the end of the incubation period, the pH decreased to 2.0, and the OD_600_ value was 6.9. Notably, after 48 h, cell growth and biotransformation occurred, even at pH 2.5 or lower. 

### 3.3. Genome Features of Strain SJF2-1 and Comparative Genome Analysis

Using Illumina and PacBio platforms, the nucleotide sequence of the complete genome of SJF2-1 was determined with 2870.0 × genome coverage. The total size was 3,198,086 bp, with 2867 protein-coding sequences, and one chromosome and six plasmids were identified. The total GC content was 58.2%. Analysis using CheckM showed 99.50% completeness and 0% strain heterogeneity. Table 1 shows the general characteristics of the SJF2-1 genome. In the JSpeciesWS server, the highest ANIb value was 97.95% (88.32% coverage) for *G. sphaericus* NBRC 12467^T^, followed by 89.69% (73.36% coverage) for *G. oxydans* LMG1408^T^. Additionally, the dDDH values (Formula 2) were 86.1% and 41.7%, respectively. These genome-based comparisons indicated that SJF2-1 belongs to the species *G. sphaericus*, based on the species threshold values of ANIb and dDDH of 95~96% and 70%, respectively [32]. There are five unpublished genome sequences of this species in the NCBI genome database (accessed on 30 September 2022), two (NBRC 12467^T^ and LMG1414^T^) from the same type of strain, and three (Dm-14, Dm-21, and Dm-28) from the same source of apple dumpster by the same research group. Genome sequences were at the contig level. In this study, the complete genome of this species was identified for the first time, and our study is the first to analyze the genes of this species. The cumulative GC mol% and GC skew plots are shown in Figure 5.

In the RAST analysis, a comparison of orthologous proteins based on the predicted protein sequences between closely related strains showed that the genes in the genome of SJF2-1 exhibited high amino acid sequence similarities to those in NBRC 12467^T^ and Dm-14 (Figure 5). For instance, of 3294 predicted proteins in SJF2-1, 1002 and 1016 proteins represented 100% identical sequences to those in NBRC 12467^T^ and Dm-14, respectively. The latter two strains share 1492 proteins with identical sequences. In contrast, 243 proteins in SJF2-1 do not match those in strains NBRC 12467^T^ and Dm-14. For example, 43 out of 53 genes in the IGS75_01300 to_1565 gene cluster were only present in SJF2-1 as a group of genes encoding hypothetical proteins with unknown domains (Figure 5). This gene cluster was also not found in the genome of *G. oxydans*. The plasmids contained fewer homologous genes compared to chromosomes between other *G. sphaericus* (Dm-14 and NBRC 12467^T^) and *G. oxydans* (621H and DSM 3504) strains (Figure 5).

At the genomic level, *G. sphaericus* is most closely related to *G. oxydans*; however, the genetic makeup and homologous gene sequence similarities between the two species differed significantly. For example, in RAST analysis, only 180 genes showed more than 99% homology with *G. oxydans* 621H^T^ [33], and 764 genes were present only in SJF2-1. Thus, SJF2-1 possesses fermentative properties that are different from those of *G. oxydans*, which are widely used in industrial applications. As a representative of this species, the genome of SJF2-1 was registered in the KEGG database (accession no. T07169), which will aid in metabolic studies of this species.

### 3.4. Genome Mining for Genes Related to Biotransformation of Glucose to 2,5-DKG

#### 3.4.1. Genes Related to PQQ (Pyrroloquinoline Quinone)-Dependent Glucose Dehydrogenase

During the oxidation of glucose to 2,5-DKG, the oxidation of D-glucose to D-gluconate by *Gluconobacter* is catalyzed by membrane-bound PQQ-dependent dehydrogenase, which first forms D-glucono-1,5-lactone from D-glucose [34]. This enzyme is a monomeric protein (approximately 87 kDa) that requires PQQ and a divalent cation, such as Mg^2+^, for the reconstitution of the active enzyme from the apoenzyme [35]. Thus, the D-glucono-1,5-lactone formed is converted to D-gluconate either spontaneously or by a membrane-bound lactonase [36]. In *G. oxydans* 621H, the GOX0265 gene (808 aa residues) encodes a PQQ-dependent glucose dehydrogenase [37]. As a result of searching for the homologous genes in the genome of SJF2-1, the IGS75_01835 gene (808 aa residues) showed the most homologous sequence to the GOX0265 gene, showing 97.5% identity in the deduced amino acid sequence (Figure 6). Both proteins have five transmembrane domains in the N-terminal region. In the genome of SJF2-1, IGS75_03370 was also found as a quinoprotein, with the same membrane topology and 37.4% amino acid sequence identity as GOX0265. These two enzymes belong to the dehydrogenases of the glucose/quinate/shikimate family. 

Other genes encoding quinoprotein dehydrogenases include two methanol/ethanol family dehydrogenases, IGS75_06000 (757 aa) and IGS75_08895 (693 aa), and glycerol/sorbitol dehydrogenase IGS75_05190 (741 aa) in the SJF2-1 genome. The genes IGS75_06000 and IGS75_08895 were accompanied by genes encoding small subunits. No transmembrane domains were identified in these three proteins. In SignalP 5.0 analysis, they possess a Sec/SPI signal peptide, indicating that the quinoprotein dehydrogenases encoded in these three genes have different topologies and membrane localizations from the two quinoprotein dehydrogenases mentioned above. 

#### 3.4.2. Genes Related to Membrane-Bound Gluconate 2-Dehydrogenase

D-gluconate is converted to 2-KG by membrane-bound gluconate 2-dehydrogenase. This enzyme has been purified and characterized from *G. dioxyacetonicus* IFO 3271, and is composed of three subunits: the FAD covalently linked large (flavoprotein) subunit (ca. 64 kDa), the cytochrome c subunit (ca. 45 kDa), and the small subunit (ca. 21 kDa) [38]. *G. oxydans* 621H is encoded by a gene cluster of *gox1232* (small subunit, 236 aa)-*gox1231* (flavoprotein subunit, 592 aa)-*gox1230* (cytochrome c subunit, 437 aa) [39]. A search for homologous genes in the SJF2-1 genome based on the amino acid sequence of GOX1231 revealed that IGS75_07935 was most homologous to GOX1232, at 79.3% identity, followed by IGS75_00745, at 59.8% identity. These genes were included in the gene clusters IGS75_07930 (210 aa)-IGS75_07935 (592 aa)-IGS75_07940 (442 aa) and IGS75_00750 (242 aa)-IGS75_00745 (592 aa)-IGS75_00740 (435 aa), respectively. The identities of the deduced amino acid sequences encoded by these genes compared with those of *G. oxydans* are shown in Figure 6. In SignalP 5.0 analysis, the Tat/SPI signal peptide was observed in small subunits (IGS75_07930 and IGS75_00750), and the Sec/SPI signal peptide was observed in cytochrome c subunits (IGS75_07940 and IGS75_00740). No signal sequence by SignalP 5.0 analysis was observed in the flavoprotein subunits (IGS75_07935 and IGS75_00745). No transmembrane domains were observed in any of the three subunits. The cytochrome c subunits were found to contain three heme c-binding CXXCH sequence motifs [40]. The large flavoprotein subunits contain GAGWAG and GFGWVG residues in the N-terminal region, corresponding to the FAD-binding motif, GXGXXG [41]. The subunits of gluconate-2-dehydrogenase in *G. oxydans* 621H were also found to possess the same signal peptides, cofactor-binding motifs, and no transmembrane domains. 

In addition, IGS75_07930-IGS75_07935-IGS75_07940 showed 96.0%-99.5%-95.2% amino acid sequence identities with those encoded in the respective *gndXYZ*, which showed the highest activity among three isofunctional gluconate 2-dehydrogenases in *G. oxydans* NBRC 3293 [42]. The high amino acid sequence identities between the proteins in these gene clusters indicate that IGS75_07930-IGS75_07935- IGS75_07940 is most responsible for functioning as gluconate 2-dehydrogenase.

#### 3.4.3. Genes Related to the Membrane-Bound 2-KG Dehydrogenase 

The reaction of 2-KG to 2,5-DKG is catalyzed by 2-KG dehydrogenase, which consists of three membrane-bound subunits, similar to gluconate 2-dehydrogenase [38]. In the case of *G. oxydans* NBRC3293, the *kgdABC* (synonym *kgdSLC*) cluster encodes a small subunit (flavoprotein)-cytochrome c subunit with 222 aa, 552 aa, and 476 aa residues, respectively [12]. In the SJF2-1 genome, the IGS75_07910 (201 aa)-IGS75_07905 (552 aa)-IGS75_07900 (476 aa) gene cluster was most homologous to *kgdABC*. The identity of the amino acid sequence between each subunit is shown in Figure 6. In SignalP 5.0 analysis, three subunits from these two gene clusters showed the same results as the membrane-bound gluconate 2-dehydrogenase. However, cytochrome c subunits contain one transmembrane domain at the C-terminus, which is not observed in membrane-bound gluconate 2-dehydrogenase.

In the SJF2-1 strain, there were three more gene clusters homologous to the *kgdABC* genes, and the identity of the amino acid sequence of each subunit is shown in Figure 6. Unusually, in the gene cluster IGS75_00810, the cytochrome c subunit gene precedes the small subunit gene. The gene clusters containing IGS75_13945 and IGS75_14610 were located in the pSJF2-a and pSJF2-b plasmids, respectively, which may have caused differences in expression due to the gene copy number.

#### 3.4.4. Phylogeny of the Membrane-Bound Flavoprotein-Cytochrome c Complexed Dehydrogenases

Analysis of the phylogeny according to the amino acid sequence of the subunits comprising D-gluconate 2-dehydrogenase and 2-keto-D-gluconate dehydrogenase families showed that each subunit is clustered into a separate clade according to two enzyme branches, as shown in Figure 7. Subunits with the same function maintained comparable distances between gene clusters. Previous studies have also shown that large subunits can be phylogenetically distinguished according to their two enzyme branches [43]. Compared to the phylogenetic distances for flavoprotein and cytochrome c subunits, longer distances between small subunits were identified, which was reflected by the low sequence similarity between them. Thus, it was predicted that the small subunits would contribute more to the stabilization of the protein structure than to the catalyst function. To date, the protein structures of these enzymes have not yet been elucidated.

## 4. Conclusions

The genus *Gluconobacter*, belonging to the acetic acid bacteria, is capable of performing oxidative fermentation with various types of sugars or alcohols. They express multiple dehydrogenases with high activity in the periplasmic space, and intracellularly for oxidation. Owing to incomplete oxidation, the conversion from the substrate to the cell mass is relatively small, yielding a high accumulation of the product. Therefore, *G. oxydans* is used in the production of industrial chemicals.

There have been no studies on biotransformation using *G. sphaericus*. The yield of D-glucose biotransformation to 2,5-DKG by *G. sphaericus* SJF2-1 was approximately 52.4% with 5% D-glucose during 96-h fermentation in flask culture. The final product is relatively homogenous. Compared with other studies with different species, our results are considered to be one of the highest yields in batch fermentation. The biotransformation efficiency could be greatly improved in the fermenter under controlled culture conditions. In addition, by using recombinant strains overexpressing the rate-limiting membrane-associated gluconate 2-dehydrogenase and mutants lacking intracellular catabolism of the intermediates, the yield may be further enhanced. For more applications of SJF2-1, mutants defective in membrane-associated gluconate 2-dehydrogenase or 2-KG dehydrogenase could be used for the production of industrially valuable D-gluconate and 2-KG. 

## Figures and Tables

**Figure 1 microorganisms-10-02130-f001:**
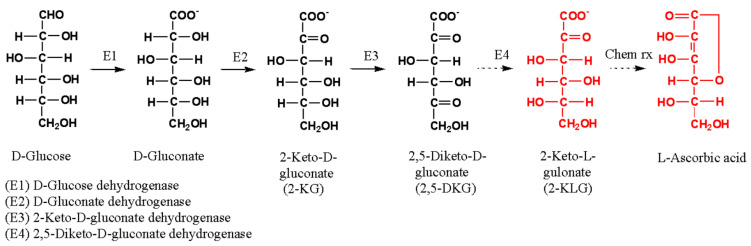
Biotransformation of D-glucose to 2-KLG via formation of 2,5-DKG, and synthesis of L-ascorbate from 2-KLG by chemical reactions.

**Figure 2 microorganisms-10-02130-f002:**
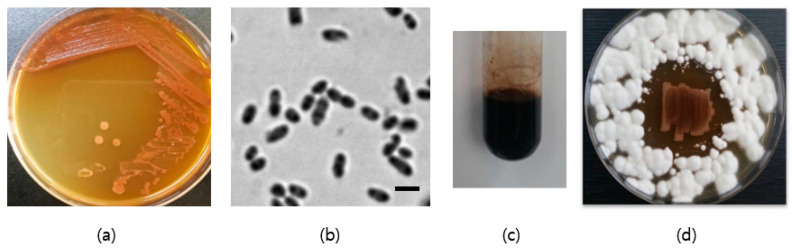
Culture properties of SJF2-1. (**a**) Production of a diffusible light brown pigment by SJF2-1 grown on YPD agar. (**b**) Cells under a phase-contrast microscope. Bar, 1 µm. (**c**) A black-colored product formed by autoclaving the YPD culture solution of SJF2-1. (**d**) Growth inhibition of *Fusarium moniliforme* var. *subglutinans* (white mycelia) by SJF2-1 on YPD agar.

**Figure 3 microorganisms-10-02130-f003:**
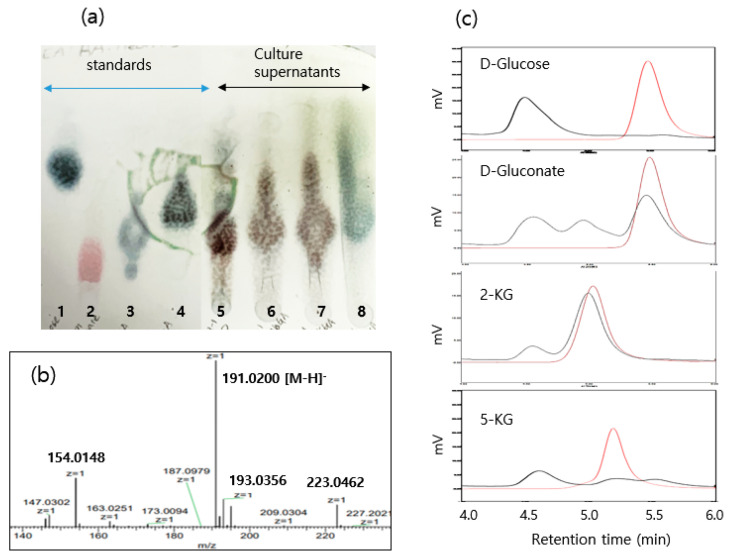
Analyses of D-glucose and sugar acid formed by *Gluconobacter sphaericus* SJF2-1. (**a**) TLC analysis (1: D-glucose 40 µg, 2: D-gluconate 400 µg, 3: 2-KG 40 µg, 4: 5-KG 40 µg, 5: YPP + 1% D-glucose, 6: YPP + 1% D-gluconate, 7: YPP + 1% 2-KG, 8: YPP + 1% 5-KG); (**b**) LC/MS analysis of product formed by D-glucose; (**c**) HPLC chromatograms of standards in red and culture supernatants in black with indicated substrate.

**Figure 4 microorganisms-10-02130-f004:**
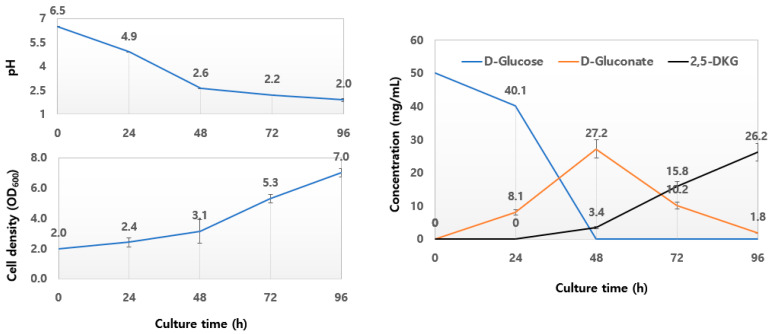
Time course of D-glucose fermentation by *Gluconobacter sphaericus* SJF2-1. Cells were cultured in 50 mL of YPP + 5% D-glucose medium as described in Materials and Methods. Sampling times are indicated. The results are from three independent experiments.

**Figure 5 microorganisms-10-02130-f005:**
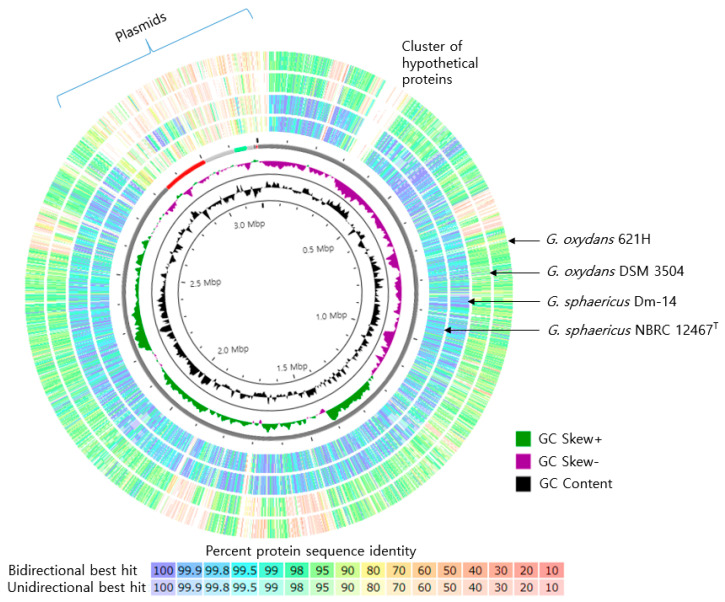
Genome comparison of *Gluconobacter sphaericus* SJF2-1 (reference) with indicated *Gluconobacter* strains conducted using RAST (outside tracks), and GC skew and GC content plots of SJF2-1 (inside). Tracks from the *Gluconobacter* strains represent pairwise BLAST comparisons between the ORFs of each strain to the ORFs of the SJF2-1 genome, with percentages of identity represented with different colors shown in the legend.

**Figure 6 microorganisms-10-02130-f006:**
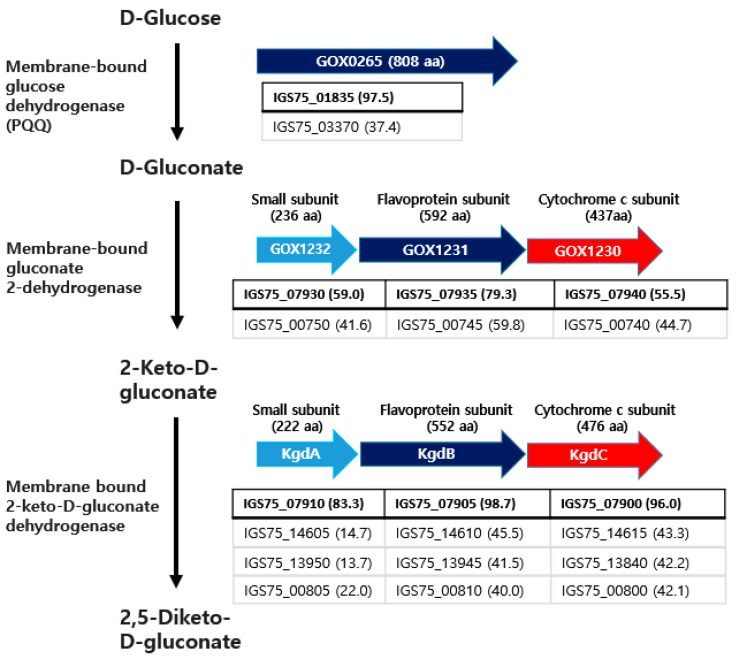
D-glucose oxidation pathway to 2,5-DKG and homologous genes encoding enzymes of the catalytic steps by SJF2-1. The genes in the arrows have been functionally demonstrated in different species, and the SJF2-1 genes in the boxes are homologous to the respective genes in the arrows. Gene clusters with the highest homology are shown in bold. Numbers in parentheses indicate the percentage of amino acid sequence identity compared to the genes in the arrows.

**Figure 7 microorganisms-10-02130-f007:**
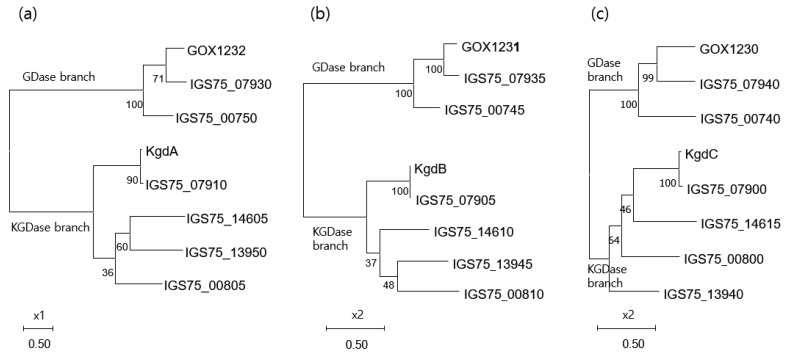
Rooted phylogenic tree of the subunits of the membrane-associated flavoprotein-cytochrome c complexed dehydrogenases using the Maximum likelihood algorithm. Trees are scaled by the amino acid distance between proteins. Bootstrap values with 1000 rapid bootstrap replicates are shown. The protein IDs of the gene clusters are shown. (**a**) Small subunit; (**b**) flavoprotein (large subunit); (**c**) cytochrome c subunit. GDase and KGDase for gluconate 2-dehydrogenase and 2-keto-D-gluconate dehydrogenase families, respectively.

**Table 1 microorganisms-10-02130-t001:** General features of the *Gluconobacter sphaericus* SJF2-1 genome.

Genome Name	GenBank Accession	Genome Size (bp)	GC (%)	No. of Coding Sequence	No. of rRNA	No. of tRNA	No. of Gene	No. of Pseudogene
Chromosome	CP068419.1	2,844,550	58.5	2518	12	56	2693	103
pSJF2-a	CP068420.1	162,291	55.7	177	- *	-	193	16
pSJF2-b	CP068421.1	109,107	55.2	102	-	-	115	13
pSJF2-c	CP068422.1	44,465	53.0	31	-	-	39	8
pSJF2-d	CP068423.1	22,858	49.3	19	-	-	29	10
pSJF2-e	CP068424.1	9589	52.0	13	-	-	13	-
pSJF2-f	CP068425.1	5226	56.3	7	-	-	7	-

* Not present.

## Data Availability

The assembled genome sequence of *G. sphaericus* SJF2-1 was deposited in GenBank under the accession numbers CP068419–CP068425 for chromosomes and plasmids. Raw sequence data used for assembly were deposited in GenBank under SRA accession numbers SRX9780842 (Illumina seq) and SRX9198931 (PacBio RSII).

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
