# Peer review of "2,5-Diketo-D-Gluconate Hyperproducing Gluconobacter sphaericus SJF2-1 with Reporting Multiple Genes Encoding the Membrane-Associated Flavoprotein-Cytochrome c Complexed Dehydrogenases"

_microorganisms, 2022, doi:10.3390/microorganisms10112130_

Round 1
Reviewer 1 Report
93 – 94: ‘For species identification of the isolate, the 16S r-RNA gene was amplified by PCR using universal primers as previously described’ - please describe in detail the methodology and applied primers, and not just give the citation.
116 – 118: I do not understand how the standard curve was prepared. Please describe more precisely.
Figure 2b is of poor quality
Author Response
We greatly appreciate valuable suggestions. The followings are our descriptions of the changes that have been made to address the concerns.
- 93 – 94: ‘For species identification of the isolate, the 16S r-RNA gene was amplified by PCR using universal primers as previously described’ - please describe in detail the methodology and applied primers, and not just give the citation.
Response 1: In the revision, the details for the experiment have been added as follows.
(revised lines 94-101). For species identification of the isolate, the 16S rRNA gene was amplified by PCR with primers 27F (5'-AGAGTTTGATCCTGGCTCAG-3') and 1492R (5'-GGTTACCTTGTTACGACTT-3'), and the nucleotide sequence obtained by Sanger sequencing was used for homology searches in the NCBI database. PCR was performed using a TRIO thermal cycler (Biometra, Germany). The amplification conditions were as follows: (1) at 95 °C for 5 min; (2) 30 cycles of 95 °C for 30 s, 53 °C for 30 s, and 72 °C for 1 min; and (3) 72 °C for 5 min. Sanger sequencing was performed at GenoTech Co. (Daejeon, Korea).
- 116 – 118: I do not understand how the standard curve was prepared. Please describe more precisely.
Response 2: In the revision, the details for the experiment have been added as follows.
(revised lines 120-126) However, 2,5-DKG is not commercially available. Because the response of the RI detector is less sensitive to sugar isomers [18], a calibration curve of 2-KG was used to roughly quantify 2,5-DKG. The calibration curve was obtained by plotting the concentrations of 2-KG (0.2%, 0.6%, 0.8%, 1.0%, and 1.2%) versus the area of the respective peaks using Microsoft Excel program (version 2019). The curve yielded a straight-line relationship, with a coefficient of determination (R2) of 0.998.
- Figure 2b is of poor quality
Response 3: The resolutions of Figure 2b have been increased.
The revised manuscript is attached.
Thank you again.

Reviewer 2 Report
A strain producing 2,5-DKG was screened by the author, and its ability to ferment and synthesize 2,5-DKG was confirmed through a series of experiments. Finally, genome sequencing was used to analyze the key genes in the genome of the strain, providing insights for the strain to be used in the synthesis of other compounds. There are some problems in the article that need to be modified.
1. The first paragraph of the summary introduces too much L-ascorbic acid. It is recommended to simplify it.
2. SJF2-1 does not significantly inhibit the growth of Fusarium moniliform var. subglutinans in Fig. 1d, and SJF2-1 may increase the inhibition effect if it occupies more area. In addition, Figure 2b is a little blurry, so it is recommended to replace the clearer picture.
3. In line 184-185, please confirm whether p.citrea has been used in this study.
4. In line 200-201, the conclusion "D-glucose, D-glucose, and 2-KG produced the same product, 2,5-DKG" has no sufficient evidence, because fig.3a lacks a positive control.
5. Fig. 3c lacks HPLC chromatograms of product 2,5-DKG, how to determine that the product in the supernatant is 2,5-DKG; How did the conclusion "The amount of 2,5-DKG was approximately 6.8 mg/mL (68% yield)" in line 215 come to?
6. The author did not describe Fig G. appearing in 5 The relationship between oxydans DSM 3504 strain and SJF2-1; In addition, NBRC 12467T in fig. 5 is inconsistent with the description in the text. Please carefully check the name of the strain in the text.
7. Results 3.5.4 is redundant in this study and cannot explain any problems, so it is recommended to delete it.
8. Please supplement the complete reference 11 and carefully check the format of other references.
Author Response
We greatly appreciate valuable suggestions. The followings are our descriptions of the changes that have been made to address the concerns.
- The first paragraph of the summary introduces too much L-ascorbic acid. It is recommended to simplify it.
Response 1: (revised lines 28-31) We deleted 2nd and 3rd sentences in the original manuscript.
- SJF2-1 does not significantly inhibit the growth of Fusarium moniliform var. subglutinans in Fig. 1d, and SJF2-1 may increase the inhibition effect if it occupies more area. In addition, Figure 2b is a little blurry, so it is recommended to replace the clearer picture.
Response 2: We replaced Fig.2d with a picture showing more area of SJF2-1 growth and significant growth inhibition of the fungus, Fusarium moniliform var. subglutinans. We replaced to Figure 2b with a high resolution.
- In line 184-185, please confirm whether p.citrea has been used in this study.
Response 3: We did not use the strain in our study. The sentence was changed the sentence as follows. (revised lines, 191-193) This phenomenon was also reported in Pantoea citrea, which is known to cause pink disease in pineapples [31]. In this study, the black color was caused by 2,5-DKG produced by P. citrea.
- In line 200-201, the conclusion "D-glucose, D-glucose, and 2-KG produced the same product, 2,5-DKG" has no sufficient evidence, because fig.3a lacks a positive control.
Response 4: We concluded the metabolite identification as 2,5-DKG based on LC/MS data and previous results based on P. citrea. Our writing in the original manuscript was too strong to reach the conclusion. In the revision, we changed as follows. (revised lines, 207-211) The substrates D-glucose, D-gluconate, and 2-KG produced a single dominant product with the same color. The supernatant of D-glucose was analyzed by LC/MS, and the largest mass peak with an m/z of 191.02 [M-H+], expected for 2,5-DKG, was identified (Figure 2(b)). Thus, the products produced from D-gluconate, and 2-KG were concluded to be 2,5-DKG.
- Fig. 3c lacks HPLC chromatograms of product 2,5-DKG, how to determine that the product in the supernatant is 2,5-DKG; How did the conclusion "The amount of 2,5-DKG was approximately 6.8 mg/mL (68% yield)" in line 215 come to?
Response 5: The calculation is based on the amount of 2-ketogluconate. The reason and the details for the calculation were described in the revised manuscript (Materials and Methods) as below. Thus, we put ‘approximately’ of ‘roughly‘ for the description of the product yield. Without the authentic compound (unfortunately 2,5-DKG is not commercially available), any method to evaluate the concentration of 2,5-DKG could have error potential.
(revised lines, 120-126) However, 2,5-DKG is not commercially available. Because the response of the RI detector is less sensitive to sugar isomers [18], a calibration curve of 2-KG was used to roughly quantify 2,5-DKG. The calibration curve was obtained by plotting the concentrations of 2-KG (0.2%, 0.6%, 0.8%, 1.0%, and 1.2%) versus the area of the respective peaks using Microsoft Excel program (version 2019). The curve yielded a straight-line relationship, with a coefficient of determination (R2) of 0.998.
- The author did not describe Fig G. appearing in 5 The relationship between oxydans DSM 3504 strain and SJF2-1; In addition, NBRC 12467T in fig. 5 is inconsistent with the description in the text. Please carefully check the name of the strain in the text.
Response 6: We put strain DSM 3504 for comparison in the genome map in Figure 5, and the description of the strain was included in text as follow.
(revised lines, 275-277) The plasmids contained fewer homologous genes compared to chromosomes between other G. sphaericus (Dm-14 and NBRC 12467T) and G. oxydans (621H and DSM 3504) strains (Figure 5). In addition, we put “T” on 12467 and ‘Plasmids’ instead of ‘plasmids’ in the revised Figure 5. However, we could not find any inconsistency between Fig. 5 and descriptions about NBRC 12467T in the text.
- Results 3.5.4 is redundant in this study and cannot explain any problems, so it is recommended to delete it.
Response 7: In this section, we showed for the first time the phylogenic relationship between gluconate 2-dehydrogenase and 2-keto-D-gluconate dehydrogenase. The result also showed phylogenetic relationship between subunits. Thus, we think the section 3.5.4 (now 3.4.4) provides novel information that cannot be covered by the result in section 3.4.3. In addition, this journal “MICROORGANISM” does not limit the amount of result description. Thus, we would like to keep the section 3.4.4.
- Please supplement the complete reference 11 and carefully check the format of other references.
Response 8: We checked all references and corrected them to the journal format.
The revised manuscript is attached.
Thank you again.

Reviewer 3 Report
The study is well designed and the manuscript is easy to read. I suggest the author include more background to talk about why some strains Erwinia and Gluconobacter natually produce/accumulate/secret 2,5-DKG. Is it an end product (which seems nonsense as microbes typically do not waste glucose to produce useless products) or a precursor to something else? This is relevant since the native pathway(s) downstream 2,5-DKG could become competing pathways if these chassis strains are engineered for biotechnological applications in the future, such as to produce ascorbic acid in these strains. One minor error -- "Figure 1" from 165 - 189 should have been Figure 2.
Author Response
We greatly appreciate valuable suggestions. The followings are our descriptions of the changes that have been made to address the concerns.
The study is well designed and the manuscript is easy to read. I suggest the author include more background to talk about why some strains Erwinia and Gluconobacter natually produce/accumulate/secret 2,5-DKG. Is it an end product (which seems nonsense as microbes typically do not waste glucose to produce useless products) or a precursor to something else? This is relevant since the native pathway(s) downstream 2,5-DKG could become competing pathways if these chassis strains are engineered for biotechnological applications in the future, such as to produce ascorbic acid in these strains. One minor error -- "Figure 1" from 165 - 189 should have been Figure
Response 1: The biotransformation from D-glucose to 2,5-DKG is an oxidative pathway yielding six electrons, which will be used for energy production as described in the Introduction. We think the pathway is like ethanol production by yeast. In this case, ethanol formed is almost the end product. However, it has not been known about the function of 2,5-DKG and further metabolism of 2,5-DKG by other bacteria. Strong acid production by 2,5-DKG can be profitable for acid-tolerant bacteria in competition under high glucose concentrations as we have shown in Fig. 2. One paper reported degradation of 2,5-DKG in the highly acidic condition. Further studies are required to find the role and further transformation of 2,5-DKG.
Response 2: We corrected the Figure number in the revised manuscript.
The revised manuscript is attached.
Thank you again.
